behaviour/cognition

optimal foraging theory, foraging cognition, priming effect, state-dependent valuation, reinforcement learning

**Author for correspondence:**
Chuan-Chin Chiao
e-mail: ccchiao@life.nthu.edu.tw

# Learned valuation during forage decision-making in cuttlefish

Tzu-Hsin Kuo[1,2] and Chuan-Chin Chiao[1,2]

[1]Institute of Systems Neuroscience, National Tsing Hua University, 101, Section 2, Kuang-Fu Road, Hsinchu 30013, Taiwan
[2]Department of Life Science, National Tsing Hua University, Hsinchu, Taiwan

 C-CC, 0000-0001-9506-0230

Decision-making, when humans and other animals choose between two options, is not always based on the absolute values of the options but can also depend on their relative values. The present study examines whether decision-making by cuttlefish is dependent on relative values learned from previous experience. Cuttlefish preferred a larger quantity when making a choice between one or two shrimps (1 versus 2) during a two-alternative forced choice. However, after cuttlefish were primed under conditions where they were given a small reward for choosing one shrimp in a no shrimp versus one shrimp test (0 versus 1) six times in a row, they chose one shrimp significantly more frequently in the 1 versus 2 test. This reversed preference for a smaller quantity was not due to satiation at the time of decision-making, as cuttlefish fed a small shrimp six times without any choice test prior to the experiment still preferred two shrimps significantly more often in a subsequent 1 versus 2 test. This suggests that the preference of one shrimp in the quantity comparison test occurs via a process of learned valuation. Foraging preference in cuttlefish thus depends on the relative value of previous prey choices.

## 1. Introduction

Optimal foraging theory, a model commonly used to study how an animal behaves when searching for food, predicts that an animal gains the most benefit from food that has the lowest cost during foraging; this approach can maximize fitness. However, animals might also optimize their foraging strategy via more complex mechanisms including enhanced spatial memory, value-based decision-making and executive control when exploiting food resources [1]. In traditional animal foraging studies, it is often assumed that value of food sources is associated with food quality and cost in time or energy needed to acquire the food. This assumption predicts that food preference of animals occurs via

learning how to maximize the net gain and minimize the net cost of the food [2]. However, this classical learning model does not address the subject's cognitive state, and recent studies have suggested that an animal's past experiences or conditions at the time of learning, rather than the expected benefit, can dominate when an animal shows prey preference [3,4]. For example, it has been shown that pigeons appear to place an increased value on the place where the food is found even if they must make more effort to obtain food at a particular location [5]. Another food-choice bias that is associated with the so-called 'sunk cost' of food also occurs in animals. One such example is where pigeons chose to stay within the situation into which they have made their initial investment, even when they have to peck more to obtain food [6]. In addition, the 'context-dependent utility' framework indicates that animals can make a choice not only based on the utility of the final outcome but also depending on both the subject's state and the background context. For example, grey jays that initially were able to obtain a large reward for a choice would devalue the same reward during a subsequent choice because of perceived danger [7].

Cuttlefish are opportunistic predators and they prey upon a variety of crustaceans and fish [8]. Cuttlefish are also known to have the most complex brains among invertebrates, and thus they are able to undertake a wide range of sophisticated cognitive behaviours [9,10]. Previous studies have shown that cuttlefish are able to adapt during food choice, and that learning plays an important role in shaping their foraging behaviours. For example, cuttlefish have been shown to be able to alter their food preference based on their embryonic visual experience [11,12]. They have also been shown to flexibly change their food preference if their preferred food is devalued by coating it with quinine, which makes it bitter [13]. Moreover, when cuttlefish are faced with a trade-off between one large shrimp and two small shrimps, they choose the single larger shrimp when they feel hungry but choose the two smaller shrimps when they are satiated [14]. This study demonstrates that cuttlefish carry out state-dependent valuation when making foraging decisions. Furthermore, it has also been shown that cuttlefish are able to integrate the 'what', 'where' and 'when' components of a single event during an experiment, which is evidence of having episodic-like memory [15]. In a recent study, it has been reported that cuttlefish adopt flexible foraging strategies depending on their preferred food predictability, thus showing future-dependent foraging cognition [16].

In the present study, we investigate whether cuttlefish are capable of flexible decision-making based on a learned valuation of food by testing whether they can adjust their prey quantity preference in response to previous priming conditions. Specifically, we manipulated cuttlefish to prefer one shrimp in a choice between one and two shrimps (the 1 versus 2 condition) by increasing the food value of one shrimp using a priming effect. This reference-dependent evaluation may provide cuttlefish with a more optimal foraging strategy when exploiting food resources via value-based decision-making.

# 2. Material and methods

## 2.1. Animals

The eggs of pharaoh cuttlefish (*Sepia pharaonis*), which were spawned by wild caught females, were reared by the Aquatic Biotech Company Ltd (Yilan, Taiwan) during February 2018 and February 2019. They were then transported to National Tsing Hua University (Hsinchu, Taiwan). After hatching, the juvenile cuttlefish were housed individually in porous containers floating inside the rearing tank. Depending on their mantle length (ML), different containers were used (ML < 2 cm, kept in a container $16 \times 11 \times 6$ cm; ML > 2 cm, kept in a container $24 \times 16 \times 6$ cm). They were fed post-larvae white shrimp (*Litopenaeus vannamei*) and freshwater shrimp (*Neocaridina denticulate*) at least twice per day. When individual animals died during the experiment, cuttlefish of the same age were introduced as a replacement. In total, 55 cuttlefish were used in the present study. All procedures were approved by the Institutional Animal Care and Use Committee of the National Tsing Hua University (protocol no. 108033).

## 2.2. Aquarium system

The animals were reared in the laboratory using two closed recirculating aquaculture systems (700 l each) that were maintained at approximately 24°C and a salinity of 33 parts per thousand. Seawater in the rearing tank overflowed into the filter tank, then went through filter sponges (as a physical filter), a skimmer (to remove organic compounds), coral sand (as a bio-filter) and UV light (to kill any microorganisms present). A water pump finally returned the seawater back into the rearing tank. The photoperiod of the recirculating aquaculture systems was a 12/12 h light/dark cycle.

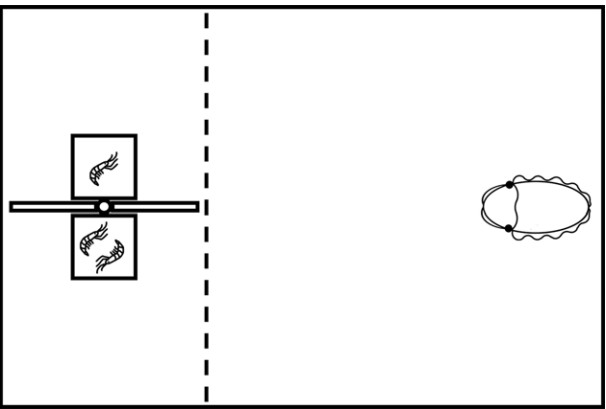

**Figure 1.** Schematic representation of the top view of the experimental set-up. The two chambers containing shrimps are placed in front of the cuttlefish, and the animal is motivated to swim toward one of the two chambers. The imaginary dotted line indicates the decision point, where the cuttlefish determines the choice.

## 2.3. Experimental apparatus and procedure

The experimental apparatus consisted of a two-chamber device made up of two small transparent plastic boxes ($2.5 \times 2.5 \times 2.5$ cm) separated by a protruding plastic sheet designed to force the cuttlefish to make an irrevocable choice (the two-alternative forced choice design, or 2AFC; figure 1). Depending on the experiment, different numbers of shrimps were placed in two chambers and the apparatus was lowered into the floating container at the opposite side from the cuttlefish's location. To prevent the startle behaviour when the experimental apparatus was lowered into the home tank, cuttlefish were habituated to the presence of apparatus and the lowering process prior to testing. The experiment was initiated by allowing the cuttlefish to swim toward the two chambers and make a choice. Cuttlefish actively prey on the live shrimps. They capture them by shooting out their two tentacles to make a strike. This behaviour is visually driven in cuttlefish [17]. Once the cuttlefish had passed an imaginary dotted line (figure 1), the apparatus was lifted out of the water to prevent passive avoidance learning, an inhibition of predatory behaviour observed during the 'prawn-in-the-tube' training procedure [18]. Except for the priming phase in *Treatment group* and *Control group II* (see below), the cuttlefish were not rewarded with any food, and this is to avoid the reinforcement of animal's decision when making a choice. In no cases cuttlefish were rewarded with the shrimp that was present in the apparatus once a choice was made. Each cuttlefish was tested six times during a trial, and each experiment contained three trials, specifically 18 tests in total, unless otherwise stated. Within each trial, the two chambers were swapped left-and-right sequentially to minimize the cuttlefish's visual lateralization effect [19]. Before each trial, the cuttlefish were forced to fast for at least 12 h. After cuttlefish had completed the six tests within a trial, they were fed an adequate amount of food in the absence of the apparatus. To ensure that shrimps were vigorously active during the experiment, the chambers were refreshed every 10–15 min. The tops of the chambers were marked clearly with the numbers of shrimp inside, and a digital video camera (Olympus STYLUS TG-5) was mounted above to record the responses of the cuttlefish. All experiments were conducted during daytime (9.00 to 21.00) and in the home tank of the animals.

## 2.4. Experimental design

### 2.4.1. Treatment group

To investigate whether priming can affect the cuttlefish's foraging decision, a series of prey quantity preference tests were carried out. Live shrimps of roughly 50% of the cuttlefish's body length were used as the prey options in the 2AFC test, and live tiny shrimps of roughly 20% of the cuttlefish's body length were used as rewards in this experiment. This experiment consisted of the priming phase and the testing phase. During the priming phase, the choice of 0 versus 1 shrimp was first presented, and the cuttlefish were expected to choose the side with one shrimp significantly more often than the side with zero shrimp. Each time the cuttlefish chose one shrimp, they were not fed the shrimp within the chamber, instead they received one tiny shrimp as a reward. This process is designed to raise the value of 'one shrimp' in the choice test. The priming phase was completed when the cuttlefish continuously chose one shrimp for six times and received a total of six tiny shrimps in a row.

Immediately or 1 hour after the priming phase, the testing phase started. During this phase, the choice of 1 versus 2 shrimps was examined without any reward for each choice.

### 2.4.2. Control group I

To investigate whether cuttlefish naturally prefer small or large quantity of shrimp, the choice of 1 versus 2 shrimps was examined without the priming phase. This was done for each cuttlefish prior to the main experiment.

### 2.4.3. Control group II

To control for the cuttlefish's prey quantity preference at different satiation levels, cuttlefish were given six tiny shrimps without making the choice of 0 versus 1 shrimp during the priming phase and they underwent the same testing phase immediately, namely the choice of 1 versus 2 shrimps. Each cuttlefish only underwent one trial (six tests) during this control experiment.

Summary of all experiments conducted in the present study is shown in electronic supplementary material, table S1.

## 2.5. Data analysis

To check parametric assumptions, we used the Kolmogorov–Smirnov normality test. Results revealed that our data were not normally distributed ($p = 0.025, 0.013, 0.200$ and $0.030$, respectively) in all four experiments. The numbers for different choices during the 2AFC test (1 versus 2) were thus compared using the non-parametric Wilcoxon signed-rank test. All statistics were conducted using SPSS.

## 3. Results

To establish cuttlefish's natural preference regarding large quantities of shrimp [14], a control experiment with the choice of 1 versus 2 shrimps was first carried out (*Control group I*). The results showed that cuttlefish chose the side with 2 shrimps significantly more often than the side with 1 shrimp (figure 2*a*; $Z = 5.40$, $p < 0.001$, $n = 39$), which suggests that cuttlefish innately preferred a larger quantity of prey. Next, we examined whether rewarding the cuttlefish with one tiny shrimp when choosing the one shrimp side during the choice test of 0 versus 1 shrimp each time for six consecutive times influenced the cuttlefish's foraging decision in the subsequent choice test of 1 versus 2 shrimps (*Treatment group*). After this priming, cuttlefish chose the side with 1 shrimp more often than the side with 2 shrimps (figure 2*b*; $Z = 3.26$, $p = 0.001$, $n = 21$). This is the opposite response to the natural quantity choice experiment (1 versus 2 shrimps; figure 2*a*) and suggests that the priming raises the 'value' of one shrimp significantly, and that cuttlefish are able to change their foraging decisions in a flexible manner depending on their immediate prior experience. Even when the test was conducted 1 hour after the priming, the cuttlefish's natural preference regarding a large quantity was still affected, although the percentages for the choice of 1 versus 2 shrimps were not significantly different (figure 2*c*; $Z = 1.58$, $p = 0.114$, $n = 13$). This suggests that the priming effect is relatively long lasting, and that prior experience indeed has a profound influence on the cuttlefish's foraging decisions.

To rule out the possibility that the observation whereby cuttlefish chose the 1 shrimp side significantly more than the 2 shrimp side after the priming was not a result of the satiation during the priming phase (the feeding of tiny shrimps as a reward), cuttlefish, after receiving six tiny shrimps without going through the task of choosing 0 versus 1 shrimp, were given the 2AFC test of 1 versus 2 shrimps immediately after feeding (*Control group II*). The results showed that they chose the side with 2 shrimps significantly more often than the side with 1 shrimp (figure 2*d*; $Z = 2.74$, $p = 0.006$, $n = 10$). These findings indicate that cuttlefish naturally prefer a large quantity of prey even in a relatively satiated state [14] and that the priming effect has altered the value of one shrimp in a manner that is independent of the cuttlefish's metabolic state.

## 4. Discussion

Cuttlefish are opportunistic predators but their foraging strategies are flexible [16]. In the wild, the prey that cuttlefish feed on are often distributed heterogeneously and different prey types vary significantly in their

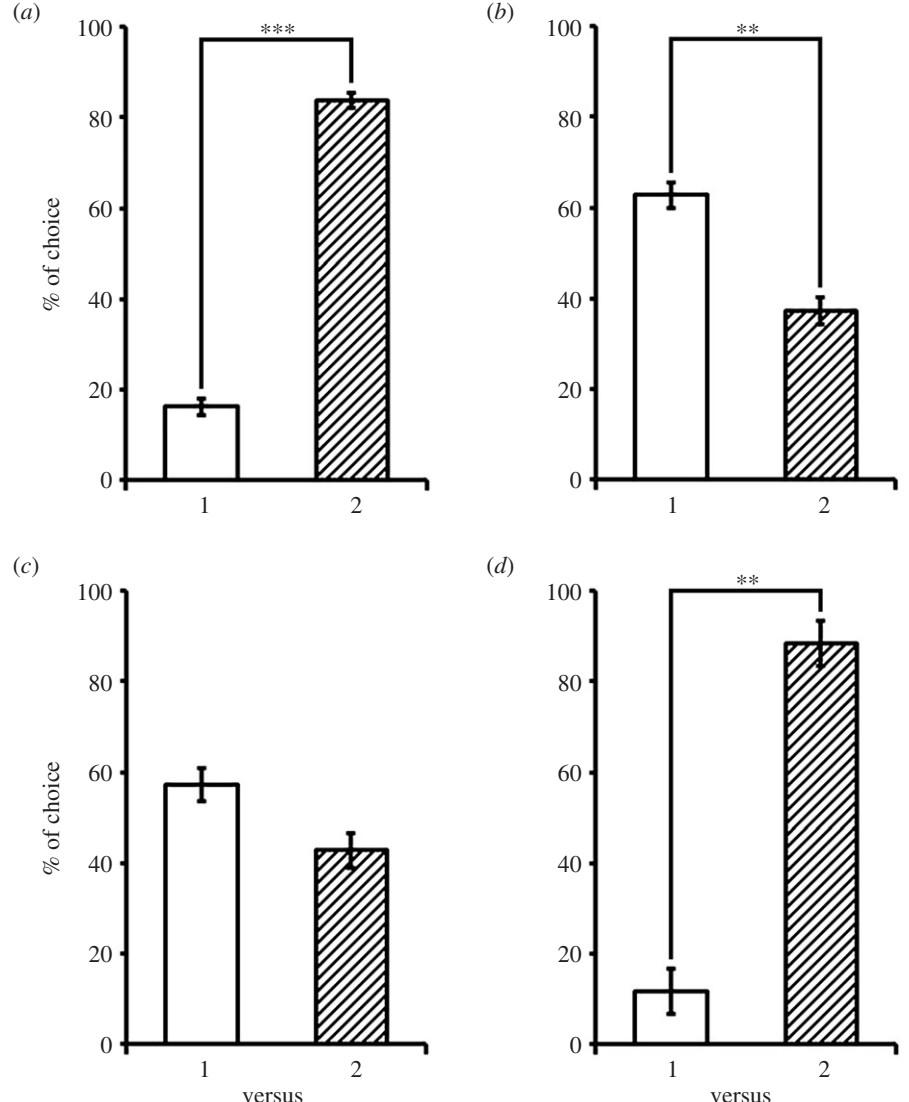

**Figure 2.** Priming of prey value alters foraging decisions of cuttlefish. Cuttlefish were subjected to the number discrimination task of 1 versus 2 shrimps in four different conditions. (a) Cuttlefish innately chose the side with 2 shrimps significantly more often than the side with 1 shrimp. (b) Immediately after priming, cuttlefish chose the side with 1 shrimp significantly more often than the side with 2 shrimps. (c) One hour after priming, the percentage choices between 1 versus 2 shrimps were not significantly different. (d) Cuttlefish without experiencing the priming effect chose the side with 2 shrimps significantly more often than the side with 1 shrimp even when cuttlefish were in a satiated state. Error bars are s.e.m. $**p < 0.01$, $***p < 0.001$.

nutritional value [8]. Thus, having effective foraging strategies is important to cuttlefish growth and survival [10]. Our observation that priming cuttlefish with a reward for choosing one shrimp in the test of 0 versus 1 is able to significantly increase the value of 'one shrimp' over 'two shrimps' in the subsequent test of 1 versus 2 (figure 2b) demonstrates that cuttlefish are able to change their foraging decisions depending on their prior experience. This phenomenon may be akin to prospect theory in behavioural economics, where decision-making is not based on absolute outcomes, but rather on relative perceptions of gains and losses [20]. This is different from the expected utility theory wherein decisions are made by assessing the expected utility of the different options and then choosing the one with the greatest utility [21]. Specifically, prospect theory asserts that the utility of each option is determined relative to a reference point, such as prevailing conditions or former experiences [22–25]. A recent study examining the food decision-making of black garden ants has shown that ants expecting to find low-quality food had a higher acceptance of medium-quality food than those expecting medium quality [4]. This indicates that relative value perception and judgement is an important element during animal decision-making.

In the present study, when cuttlefish did not experience a priming effect, they adopted the most optimal foraging strategy by choosing two shrimps more often in the 1 versus 2 test (figure 2a),

which confirms their innate preference for a larger quantity of prey [14]. However, after learning the value of one shrimp relative to the value of zero shrimp (no food) via the priming phase (*Treatment group*), the cuttlefish reversed their preference for the smaller quantity of prey in the subsequent test of 1 versus 2 shrimps (figure 2*b*), which suggests that reference-dependent evaluation may modulate their optimal foraging strategy via value-based decision-making. This effect was quite long lasting. Even 1 hour after the priming phase, cuttlefish had not completely resumed their innate preference for a larger quantity of prey (figure 2*c*). Further experiments showed that this valuation effect was probably dependent on the animals' learning processes rather than their metabolic state. When cuttlefish were fed six tiny shrimps without experiencing the priming effect, they maintained their innate preference for the larger quantity of prey in the 1 versus 2 shrimp test (figure 2*d*). The present results thus demonstrate that cuttlefish are capable of flexible decision-making based not only on the prevailing environmental conditions but also on the previous learned valuation.

Alternatively, the reversed preference of cuttlefish's food choice via the priming phase can be explained by positive reinforcement learning in an instrumental conditioning experiment. Instrumental conditioning is an important type of associative learning in which an animal learns to associate a behaviour with positive or negative reinforcement (reward or punishment) delivered after the animal has made a response [26]. It results in the animal repeating the behaviour with a greater or lower frequency. For example, in the prawn-in-the-tube test, cuttlefish soon learned to stop visually attacking the prawn, and the level of striking was inversely related to the level of negative reinforcement [27]. Previous studies have shown that prey preference of cuttlefish is modified by training, striking evidence for taste aversion and appetite learning [13,28]. Using escape as a positive reinforcement, it has also been found that cuttlefish are capable of spatial learning [29,30] and conditional discrimination [31]. The present study demonstrates that cuttlefish learned to associate the choice of one shrimp with a reward. This result suggests that cuttlefish are sensitive to positive reinforcement. Without further evidence, the parsimonious interpretation of our finding is that the observed effect was most likely caused by prior positive reinforcement during priming. Regardless of the type of learning, this reference-dependent evaluation may help to provide cuttlefish with an optimal foraging strategy when exploiting food resources.

Ethics. This work was carried out in accordance with the EU-Directive 2010/63/EU, and all procedures were approved by the Institutional Animal Care and Use Committee of the National Tsing Hua University (protocol no. 108033).
Data accessibility. Data available in the electronic supplementary material.
Authors' contributions. T.-H.K. conceived, designed, carried out the work and drafted the manuscript. C.-C.C. helped plan experiments, interpreted data and revised the manuscript.
Competing interests. We have no competing interests to declare.
Acknowledgements. We thank Tsang-I Yang, Chun-Yen Lin, Tsung-Han Liu, Yao-Chen Lee and Yung-Chieh Liu for their help with the experimental design. This study was supported by the Ministry of Science and Technology of Taiwan, MOST 106-2311-B-007-010-MY3 (to C.-C.C.).

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
