## [Reviewer comments · Royal Society Open Science]

Review History

RSOS-200917.R0 (Original submission)

Review form: Reviewer 1

Is the manuscript scientifically sound in its present form?

No

Are the interpretations and conclusions justified by the results?

No

Is the language acceptable?

No

Do you have any ethical concerns with this paper?

No

Have you any concerns about statistical analyses in this paper?

No

Recommendation?

Reject

Comments to the Author(s)

This manuscript simply shows that cuttlefish are sensitive to positive reinforcement. They repeated the actions reinforced during “priming” in the critical 1 vs. 2 tests. The authors acknowledge this in the discussion. Yet, they decided to frame the entire manuscript in cognitive and ecological terms. Given that reinforcement learning can account for the entire set of findings appealing to behavioural economics, less is more, loss aversion and the like seems inappropriate. In the discussion, the authors write “this learned valuation effect was likely to dependent on the animals' cognitive processes...”. My reading is that the effect was most likely caused by prior positive reinforcement of the actions displayed at test. Morgan's Canon may help us decide.

Review form: Reviewer 2 (Alexandra Schnell)

Is the manuscript scientifically sound in its present form?

Yes

Are the interpretations and conclusions justified by the results?

Yes

Is the language acceptable?

Yes

Do you have any ethical concerns with this paper?

No

Have you any concerns about statistical analyses in this paper?

Yes

Recommendation?

Accept with minor revision (please list in comments)

Comments to the Author(s)

RS0S-200917: Learned valuation during forage decision making in cuttlefish

This is a comprehensive and important study. The authors use an alternative force two-choice paradigm to investigate learned valuation in cuttlefish. The experiment is well-designed and the evidence for the conclusions is convincing. My main concerns outlined below can hopefully be addressed with some additional explanation in the text. The writing requires some attention as some sections are poorly worded. I have made some suggestions and provide comments that are aimed to help the authors improve the manuscript.

Specific comments

Line 49: consider revising the end of this sentence to ‘this approach can maximize fitness’.

Lines 50-52: consider revising these two sentences to form a single succinct sentence: ‘However, animals might also optimize their foraging strategy via more complex mechanisms including spatial memory, value-based decision-making, and executive control when exploiting food resources.’

Line 71: Please remove the word ‘constantly’ as this suggests that cuttlefish spend most of their time foraging. Accelerometry research has shown that cuttlefish have long rest periods that are punctuated by brief foraging bouts (Aitken, O’Dor & Jackson, 2005).

Line 79: insert the word ‘it’ i.e. ‘by coating it with quinine’.

Lines 79-84: this sentence is poorly worded, please revise and make it clear that you are referring to a different study than the previous quinine study. i.e. ‘Moreover, when cuttlefish are faced with a trade-off between one large shrimp and two small shrimp, they choose the single larger shrimp when they are hungry but choose the two smaller shrimp when they are satiated.’

Line 82: omit the word 'previous'.

Line 85: best to refer to it as 'what', 'where' and 'when' components as episodic-like memory can often be termed what-where-when memory.

Line 86: omit the word 'an' before 'episodic-like memory'. Also, omit the word 'very' before 'recent study'.

Line 87: omit the words 'are able to' and just say 'cuttlefish adopt flexible foraging...'

Line 91: omit the words 'and we did this'

Line 93: consider revising to 'Specifically, we manipulated cuttlefish to prefer one shrimp in a choice between one and two shrimp by increasing the food value of one shrimp using a priming effect'.

Line 94: a space is missing between 'the' and '1'.

Line 105: omit the word 'during' before February 2019, one 'during' is sufficient in this sentence.

Line 120: salinity is typically reported as parts per thousand (ppt) not %.

Line 123: replace the word 'drew' with 'returned'.

Line 128: methods are typically written in past tense, thus please consider changing the word 'consists' to 'consisted'.

Line 132: in my experience cuttlefish startle easily, did the cuttlefish exhibit any startle behaviours when the floating container was lowered into the home tank? Had the cuttlefish been habituated to the apparatus and lowering process prior to testing? If yes, please include this in the methods.

Lines 137-140: this sentence is a little confusing. Was the entire apparatus lifted out of the water? If yes, please include an explanation as to why the cuttlefish were not rewarded with the food within the chamber once a choice was made? I assume it is because the authors did not want to reinforce the subject's choice in this initial stage of the experiment, but this needs to be stated in the methods.

Line 143: The study by Schnell et al. 2016. Lateralization of eye use in cuttlefish, is a more appropriate reference to use here because it demonstrates visual lateralization during prey capture, whereas the Jozet-Alves et al. 2012. study demonstrates visual lateralization when searching for shelter in a two-choice arm maze.

Line 163: what exactly constitutes a tiny shrimp? Can you please include parameters?

Lines 182-185: Can the authors please provide a little more detail in the statistical section. Did the authors first conduct normality tests to check parametric assumptions? If yes, please state the type of normality test and the results of that test to justify why non-parametric tests were used.

Line 191: consider revising to '...cuttlefish innately preferred a larger quantity of prey'.

Line 196: why 'apparently'? Consider omitting this word.

Lines 220-221: this sentence is poorly worded, please consider revising. i.e. 'in the wild, the prey that cuttlefish feed on are often distributed heterogeneously and different prey types vary significantly in their nutritional value.

Line 223: omit the word 'simply'.

Line 227: omit the word 'the' before 'prospect theory'; same comment for line 231.

Line 232: consider replacing 'status quo' with 'prevailing conditions'.

Lines 232-233: due to poor wording, please consider revising the end of sentence to read: 'for instance, in reference to prevailing conditions or former experiences.'

Lines 233-236: Great example □

Line 238: please consider revising this sentence to read: '...when cuttlefish did not experience a priming effect, they adopted the...'

Line 246: consider revising sentence to read: 'the apparent loss aversion effect was prolonged'.

Line 249: omit the word 'to' before 'dependent'.

Line 250: omit the word 'with' before 'six tiny shrimp'; also consider revising sentence to read: 'when cuttlefish were fed six tiny shrimp without experiencing the priming effect,...'

Line 254: consider replacing the word 'given' with 'prevailing'

Lines 255-259: please consider revising this section as the sentence it is poorly worded. For instance, consider breaking the sentence into two parts. Please also consider defining 'instrumental conditioning experiment', as a general biological audience that does not study animal cognition might need further explanation.

Line 259: missing the word 'of' after 'regardless'

Lines 259-261: please consider revising the sentence. Currently, it's a little confusing. What exactly is helping the cuttlefish optimise their foraging strategy? Reference-dependent evaluation or value-based decision making or both?

Lines 267-270: please consider revising the last sentence. The statement requires expanding and further justification. Why is foraging cognition in cuttlefish equivalent to that of vertebrates?

What specific aspects are equivalent? Are the authors referring to the previous statement in that state-dependent valuation learning is rife across diverse taxa? If yes, then foraging cognition in cuttlefish is not only akin to vertebrates but also other invertebrates i.e. insects. If no, what specific aspects of learned valuation is vertebrate-like? Why is this important?

Lines 365-373: Please revise the figure 2 legend to clearly state that cuttlefish in the experiment depicted in 2D did not get exposed to a priming effect.

Decision letter (RSOS-200917.R0)

Dear Dr Chiao:

Manuscript ID RSOS-200917 entitled "Learned valuation during forage decision making in cuttlefish" which you submitted to Royal Society Open Science, has been reviewed. The comments from reviewers are included at the bottom of this letter.

In view of the criticisms of the reviewers, the manuscript has been rejected in its current form. However, a new manuscript may be submitted which takes into consideration these comments.

Please note that resubmitting your manuscript does not guarantee eventual acceptance, and that your resubmission will be subject to peer review before a decision is made.

Your resubmitted manuscript should be submitted by 18-Jan-2021. If you are unable to submit by this date please contact the Editorial Office.

on behalf of Dr Kimberley Mathot (Associate Editor) and Kevin Padian (Subject Editor)
openscience@royalsociety.org

Associate Editor Comments to Author (Dr Kimberley Mathot):

Comments to the Author:

I have now received two reviews of your manuscript entitled “Learned valuation during forage decision making in cuttlefish”. Referee #1 expressed concerns about the appropriateness of invoking “less is more” and “loss aversion” arguments when interpreting the results. Upon careful reading of the manuscript, I felt that the description of the methods was not sufficiently clear to allow me to assess this. Referee #2 also pointed out that several aspects of the methods are unclear. Some of the details that are unclear after reading the manuscript make it impossible to assess the soundness of the interpretations that have been drawn from the results. As such, I am recommending that the manuscript be rejected with possibility to resubmit following substantial clarification of the methods.

Should you choose to resubmit the manuscript, please address the following points, in addition to the very detailed points provided by referee #2.

1. A schematic representation of the study setup would be useful – including the treatment orders, choices provided, rewards associated with each choice, and sample sizes (number of fish) and number of trials.
2. Please be explicit about what total rewards were offered in each phase, including, for example, whether a single shrimp choice plus the additional reward of a “tiny” shrimp was equivalent to, or less than the reward experienced when choosing 2 shrimp in the baseline preference test (or were these trials all unrewarded?). This is critical for interpreting the change in preference in the third phase (preference for 1 shrimp over 2 shrimp) as the interpretation if 2 shrimp was never rewarded is very different than if 2 shrimp was rewarded with 2 shrimp in the baseline trial. If the former, then the study demonstrates associative learning, if the latter, then knowing how much the “tiny” shrimp was worth becomes critical.

Reviewers' Comments to Author:

Reviewer: 1

Comments to the Author(s)

This manuscript simply shows that cuttlefish are sensitive to positive reinforcement. They repeated the actions reinforced during “priming” in the critical 1 vs. 2 tests. The authors acknowledge this in the discussion. Yet, they decided to frame the entire manuscript in cognitive and ecological terms. Given that reinforcement learning can account for the entire set of findings appealing to behavioural economics, less is more, loss aversion and the like seems inappropriate. In the discussion, the authors write “this learned valuation effect was likely to dependent on the animals' cognitive processes...”. My reading is that the effect was most likely caused by prior positive reinforcement of the actions displayed at test. Morgan's Canon may help us decide.

Reviewer: 2

Comments to the Author(s)

RS05-200917: Learned valuation during forage decision making in cuttlefish

This is a comprehensive and important study. The authors use an alternative force two-choice paradigm to investigate learned valuation in cuttlefish. The experiment is well-designed and the evidence for the conclusions is convincing. My main concerns outlined below can hopefully be addressed with some additional explanation in the text. The writing requires some attention as some sections are poorly worded. I have made some suggestions and provide comments that are aimed to help the authors improve the manuscript.

Specific comments

Line 49: consider revising the end of this sentence to ‘this approach can maximize fitness’.

Lines 50-52: consider revising these two sentences to form a single succinct sentence: 'However, animals might also optimize their foraging strategy via more complex mechanisms including spatial memory, value-based decision-making, and executive control when exploiting food resources.'

Line 71: Please remove the word 'constantly' as this suggests that cuttlefish spend most of their time foraging. Accelerometry research has shown that cuttlefish have long rest periods that are punctuated by brief foraging bouts (Aitken, O'Dor & Jackson, 2005).

Line 79: insert the word 'it' i.e. 'by coating it with quinine'.

Lines 79-84: this sentence is poorly worded, please revise and make it clear that you are referring to a different study than the previous quinine study. i.e. 'Moreover, when cuttlefish are faced with a trade-off between one large shrimp and two small shrimp, they choose the single larger shrimp when they are hungry but choose the two smaller shrimp when they are satiated.'

Line 82: omit the word 'previous'.

Line 85: best to refer to it as 'what', 'where' and 'when' components as episodic-like memory can often be termed what-where-when memory.

Line 86: omit the word 'an' before 'episodic-like memory'. Also, omit the word 'very' before 'recent study'.

Line 87: omit the words 'are able to' and just say 'cuttlefish adopt flexible foraging...'

Line 91: omit the words 'and we did this'

Line 93: consider revising to 'Specifically, we manipulated cuttlefish to prefer one shrimp in a choice between one and two shrimp by increasing the food value of one shrimp using a priming effect'.

Line 94: a space is missing between 'the' and '1'.

Line 105: omit the word 'during' before February 2019, one 'during' is sufficient in this sentence.

Line 120: salinity is typically reported as parts per thousand (ppt) not %.

Line 123: replace the word 'drew' with 'returned'.

Line 128: methods are typically written in past tense, thus please consider changing the word 'consists' to 'consisted'.

Line 132: in my experience cuttlefish startle easily, did the cuttlefish exhibit any startle behaviours when the floating container was lowered into the home tank? Had the cuttlefish been habituated to the apparatus and lowering process prior to testing? If yes, please include this in the methods.

Lines 137-140: this sentence is a little confusing. Was the entire apparatus lifted out of the water? If yes, please include an explanation as to why the cuttlefish were not rewarded with the food within the chamber once a choice was made? I assume it is because the authors did not want to reinforce the subject's choice in this initial stage of the experiment, but this needs to be stated in the methods.

Line 143: The study by Schnell et al. 2016. Lateralization of eye use in cuttlefish, is a more appropriate reference to use here because it demonstrates visual lateralization during prey capture, whereas the Joze-Alves et al. 2012. study demonstrates visual lateralization when searching for shelter in a two-choice arm maze.

Line 163: what exactly constitutes a tiny shrimp? Can you please include parameters?

Lines 182-185: Can the authors please provide a little more detail in the statistical section. Did the authors first conduct normality tests to check parametric assumptions? If yes, please state the type of normality test and the results of that test to justify why non-parametric tests were used.

Line 191: consider revising to '...cuttlefish innately preferred a larger quantity of prey'.

Line 196: why 'apparently'? Consider omitting this word.

Lines 220-221: this sentence is poorly worded, please consider revising. i.e. 'in the wild, the prey that cuttlefish feed on are often distributed heterogeneously and different prey types vary significantly in their nutritional value.'

Line 223: omit the word 'simply'.

Line 227: omit the word 'the' before 'prospect theory'; same comment for line 231.

Line 232: consider replacing 'status quo' with 'prevailing conditions'.

Lines 232-233: due to poor wording, please consider revising the end of sentence to read: 'for instance, in reference to prevailing conditions or former experiences.'

Lines 233-236: Great example □

Line 238: please consider revising this sentence to read: '...when cuttlefish did not experience a priming effect, they adopted the...'

Line 246: consider revising sentence to read: 'the apparent loss aversion effect was prolonged'.

Line 249: omit the word 'to' before 'dependent'.

Line 250: omit the word 'with' before 'six tiny shrimp'; also consider revising sentence to read: 'when cuttlefish were fed six tiny shrimp without experiencing the priming effect...'

Line 254: consider replacing the word 'given' with 'prevailing'

Lines 255-259: please consider revising this section as the sentence it is poorly worded. For instance, consider breaking the sentence into two parts. Please also consider defining 'instrumental conditioning experiment', as a general biological audience that does not study animal cognition might need further explanation.

Line 259: missing the word 'of' after 'regardless'

Lines 259-261: please consider revising the sentence. Currently, it's a little confusing. What exactly is helping the cuttlefish optimise their foraging strategy? Reference-dependent evaluation or value-based decision making or both?

Lines 267-270: please consider revising the last sentence. The statement requires expanding and further justification. Why is foraging cognition in cuttlefish equivalent to that of vertebrates?

What specific aspects are equivalent? Are the authors referring to the previous statement in that state-dependent valuation learning is rife across diverse taxa? If yes, then foraging cognition in cuttlefish is not only akin to vertebrates but also other invertebrates i.e. insects. If no, what specific aspects of learned valuation is vertebrate-like? Why is this important?

Lines 365-373: Please revise the figure 2 legend to clearly state that cuttlefish in the experiment depicted in 2D did not get exposed to a priming effect.

Author's Response to Decision Letter for (RSOS-200917.R0)

See Appendix A.

RSOS-201602.R0

Review form: Reviewer 1

Is the manuscript scientifically sound in its present form?

Yes

Are the interpretations and conclusions justified by the results?

Yes

Is the language acceptable?

Yes

Do you have any ethical concerns with this paper?

No

Have you any concerns about statistical analyses in this paper?

No

Recommendation?

Accept with minor revision (please list in comments)

Comments to the Author(s)

The manuscript has improved substantially. This new version avoids unnecessary claims of higher cognitive processes. I still fail to understand how the results found can be interpreted using prospect theory (page 19). The fact that decision-making is based on relative rather than on absolute metrics does not help us explain how 1 shrimp is better than 2. (by the way, a phenomenon cannot be akin to a theory, lines 233-236). Positive reinforcement is indeed a good and parsimonious explanation.

The pending issue is whether the demonstration of reinforcement learning in cuttlefish is newsworthy. I do not think it is, but the science presented in the paper is solid.

Review form: Reviewer 2 (Alexandra Schnell)**Is the manuscript scientifically sound in its present form?**

Yes

Are the interpretations and conclusions justified by the results?

Yes

Is the language acceptable?

Yes

Do you have any ethical concerns with this paper?

No

Have you any concerns about statistical analyses in this paper?

No

Recommendation?

Accept with minor revision (please list in comments)

Comments to the Author(s)

Comments attached (Appendix B).

Decision letter (RSOS-201602.R0)

Dear Dr Chiao,

On behalf of the Editors, we are pleased to inform you that your Manuscript RSOS-201602 "Learned valuation during forage decision making in cuttlefish" has been accepted for publication in Royal Society Open Science subject to minor revision in accordance with the referees' reports. Please find the referees' comments along with any feedback from the Editors below my signature.

We invite you to respond to the comments and revise your manuscript. Below the referees' and Editors' comments (where applicable) we provide additional requirements. Final acceptance of

your manuscript is dependent on these requirements being met. We provide guidance below to help you prepare your revision.

Please submit your revised manuscript and required files (see below) no later than 7 days from today's (ie 09-Nov-2020) date. Note: the ScholarOne system will 'lock' if submission of the revision is attempted 7 or more days after the deadline. If you do not think you will be able to meet this deadline please contact the editorial office immediately.

on behalf of Dr Kimberley Mathot (Associate Editor) and Kevin Padian (Subject Editor)
openscience@royalsociety.org

Associate Editor Comments to Author (Dr Kimberley Mathot):

The revised manuscript has now been reviewed by two referees. Both referees were satisfied that the revisions addressed the major comments raised in the initial review. Please address the very minor edits suggested by the referees and resubmit within 7 days for final acceptance.

Reviewer comments to Author:

Reviewer: 1
Comments to the Author(s)

The manuscript has improved substantially. This new version avoids unnecessary claims of higher cognitive processes. I still fail to understand how the results found can be interpreted using prospect theory (page 19). The fact that decision-making is based on relative rather than on absolute metrics does not help us explain how 1 shrimp is better than 2. (by the way, a phenomenon cannot be akin to a theory, lines 233-236). Positive reinforcement is indeed a good and parsimonious explanation.

The pending issue is whether the demonstration of reinforcement learning in cuttlefish is newsworthy. I do not think it is, but the science presented in the paper is solid.

Reviewer: 2
Comments to the Author(s)

Comments attached

===PREPARING YOUR MANUSCRIPT===

- one version identifying all the changes that have been made (for instance, in coloured highlight, in bold text, or tracked changes);
- a 'clean' version of the new manuscript that incorporates the changes made, but does not highlight them.

 This version will be used for typesetting.

===PREPARING YOUR REVISION IN SCHOLARONE===

- If you are requesting a discretionary waiver for the article processing charge, the waiver form must be included at this step.
- If you are providing image files for potential cover images, please upload these at this step, and inform the editorial office you have done so. You must hold the copyright to any image provided.
- A copy of your point-by-point response to referees and Editors. This will expedite the preparation of your proof.

- Ensure that your data access statement meets the requirements at <https://royalsociety.org/journals/authors/author-guidelines/#data>. You should ensure that you cite the dataset in your reference list. If you have deposited data etc in the Dryad repository, please only include the 'For publication' link at this stage. You should remove the 'For review' link.
- If you are requesting an article processing charge waiver, you must select the relevant waiver option (if requesting a discretionary waiver, the form should have been uploaded at Step 3 'File upload' above).
- If you have uploaded ESM files, please ensure you follow the guidance at <https://royalsociety.org/journals/authors/author-guidelines/#supplementary-material> to include a suitable title and informative caption. An example of appropriate titling and captioning may be found at https://figshare.com/articles/Table_S2_from_Is_there_a_trade-off_between_peak_performance_and_performance_breadth_across_temperatures_for_aerobic_scope_in_teleost_fishes_/3843624.

Author's Response to Decision Letter for (RSOS-201602.R0)

See Appendix C.

Decision letter (RSOS-201602.R1)

Dear Dr Chiao,

It is a pleasure to accept your manuscript entitled "Learned valuation during forage decision making in cuttlefish" in its current form for publication in Royal Society Open Science.

on behalf of Dr Kimberley Mathot (Associate Editor) and Kevin Padian (Subject Editor)
openscience@royalsociety.org

Associate Editor Comments to Author (Dr Kimberley Mathot):

Thank you for addressing the minor edits suggested in the second round of revisions. Your manuscript has now been accepted.

Appendix A

Associate Editor Comments to Author:

Comments to the Author:

I have now received two reviews of your manuscript entitled “Learned valuation during forage decision making in cuttlefish”. Referee #1 expressed concerns about the appropriateness of invoking “less is more” and “loss aversion” arguments when interpreting the results. Upon careful reading of the manuscript, I felt that the description of the methods was not sufficiently clear to allow me to assess this. Referee #2 also pointed out that several aspects of the methods are unclear. Some of the details that are unclear after reading the manuscript make it impossible to assess the soundness of the interpretations that have been drawn from the results. As such, I am recommending that the manuscript be rejected with possibility to resubmit following substantial clarification of the methods.

Response: We agree with Referee #1. We have toned down the “less is more” and “loss aversion” arguments in the Introduction and Discussion. We have also re-written some of the Methods sections and now the description of the methods is much clear. We appreciate your help and giving us an opportunity to resubmit the manuscript. We hope that you find this revised manuscript much improved.

Should you choose to resubmit the manuscript, please address the following points, in addition to the very detailed points provided by referee #2.

Response: We have addressed the following points and all the detailed points provided by Referee #2.

1. A schematic representation of the study setup would be useful – including the treatment orders, choices provided, rewards associated with each choice, and sample sizes (number of fish) and number of trials.

Response: Thanks for the suggestion. We have included a supplementary table to show the schematic representation of the study setup.

2. Please be explicit about what total rewards were offered in each phase, including, for example, whether a single shrimp choice plus the additional reward of a “tiny” shrimp was equivalent to, or less than the reward experienced when choosing 2 shrimp in the baseline preference test (or were these trials all unrewarded?). This is critical for interpreting the change in preference in the third phase (preference for 1 shrimp over 2 shrimp) as the interpretation if 2 shrimp was never rewarded is very different than if 2 shrimp was rewarded with 2 shrimp in the baseline trial. If the former, then the study demonstrates associative learning, if the latter, then knowing how much the “tiny” shrimp was worth becomes critical.

Response: Sorry for the confusion. We have added a supplementary table to summarize the reward or no reward associated with each choice. Except for the priming phase in Treatment group and Control group II, the cuttlefish were not rewarded with any food, and this is to avoid the reinforcement of animal’s decision when making a choice. In no cases cuttlefish were rewarded with the shrimp that was present in the apparatus once a choice was made. Thus, in the case of 1 vs. 2 choice, choosing the side with two shrimps were never rewarded. Indeed, in the principle of parsimony, the present study demonstrates associative learning, though other alternative interpretations may still exist. We have modified the Discussion in the revised manuscript accordingly.

Reviewer 1

Comments to the Author(s)

This manuscript simply shows that cuttlefish are sensitive to positive reinforcement. They repeated the actions reinforced during “priming” in the critical 1 vs. 2 tests. The authors acknowledge this in the discussion. Yet, they decided to frame the entire manuscript in cognitive and ecological terms. Given that reinforcement learning can account for the entire set of findings appealing to behavioural economics, less is more, loss aversion and the like seems inappropriate. In the discussion, the authors write “this learned valuation effect was likely to dependent on the animals' cognitive processes...”. My reading is that the effect was most likely caused by prior positive reinforcement of the actions displayed at test. Morgan's Canon may help us decide.

Response: We agree with the reviewer that reinforcement learning can equally account for the result that we observed in the present study. Indeed, Morgan’s Canon (or the principle of parsimony) suggests that we should adopt the simplest explanation. Although cuttlefish have been demonstrated to equip with substantially high level of cognitive function, without further evidence we should be cautious about our interpretation in the present study. Therefore, we have toned down our claims in the revised manuscript. Despite we kept several examples of animal’s decision making which may be indicative of cognitive state in the Introduction, we strictly explained our findings in favor of reinforcement learning in the Discussion. Specifically, we have deleted the penultimate sentence of the Introduction entirely, so that the implication of behavioral economics, less is more, loss aversion, and the like are no longer present in the Introduction. Although we only provided the alternative interpretation of cognitive process in a small portion of the Discussion, so that the readers could have a different view of our findings, we carefully reminded the readers that the principle of parsimony should be adopted for the interpretation of the results at the very end of the Discussion.

Reviewer 2

This is a comprehensive and important study. The authors use an alternative force two-choice paradigm to investigate learned valuation in cuttlefish. The experiment is well-designed and the evidence for the conclusions is convincing. My main concerns outlined below can hopefully be addressed with some additional explanation in the text. The writing requires some attention as some sections are poorly worded. I have made some suggestions and provide comments that are aimed to help the authors improve the manuscript.

Response: Thanks for the nice summary and suggestions.

Specific comments

Line 49: consider revising the end of this sentence to ‘this approach can maximize fitness’.

Response: Thanks for the suggestion. We have revised the sentence.

Lines 50-52: consider revising these two sentences to form a single succinct sentence: ‘However, animals might also optimize their foraging strategy via more complex mechanisms including spatial memory, value-based decision-making, and executive control when exploiting food resources.’

Response: Thanks for the suggestion. We have revised the sentence.

Line 71: Please remove the word ‘constantly’ as this suggests that cuttlefish spend most of their time foraging. Accelerometry research has shown that cuttlefish have long rest periods that are punctuated by *brief* foraging bouts (Aitken, O’Dor & Jackson, 2005).

Response: Thanks for the suggestion. We have removed the word ‘constantly’.

Line 79: insert the word ‘it’ i.e. ‘by coating it with quinine’.

Response: Thanks for the suggestion. We have inserted the word ‘it’.

Lines 79-84: this sentence is poorly worded, please revise and make it clear that you are referring to a different study than the previous quinine study. i.e. ‘Moreover, when cuttlefish are faced with a trade-off between one large shrimp and two small shrimp, they choose the single larger shrimp when they are hungry but choose the two smaller shrimp when they are satiated.’

Response: Thanks for the suggestion. We have revised the sentence.

Line 82: omit the word ‘previous’.

Response: Thanks for the suggestion. We have omitted the word.

Line 85: best to refer to it as ‘what’, ‘where’ and ‘when’ components as episodic-like memory can often be termed what-where-when memory.

Response: Thanks for the suggestion. We have revised the sentence.

Line 86: omit the word ‘an’ before ‘episodic-like memory. Also, omit the word ‘very’ before ‘recent study’.

Response: Thanks for the suggestion. We have omitted these words.

Line 87: omit the words ‘are able to’ and just say ‘cuttlefish adopt flexible foraging...’

Response: Thanks for the suggestion. We have omitted the words.

Line 91: omit the words ‘and we did this’

Response: Thanks for the suggestion. We have omitted the words.

Line 93: consider revising to ‘Specifically, we manipulated cuttlefish to prefer one shrimp in a choice between one and two shrimp by increasing the food value of one shrimp using a priming effect’.

Response: Thanks for the suggestion. We have revised the sentence.

Line 94: a space is missing between ‘the’ and ‘1’.

Response: Thanks for the suggestion. We have added a space between ‘the’ and ‘1’.

Line 105: omit the word ‘during’ before February 2019, one ‘during’ is sufficient in this sentence.

Response: Thanks for the suggestion. We have omitted the word.

Line 120: salinity is typically reported as parts per thousand (ppt) not %.

Response: Thanks for the suggestion. We have replaced % with parts per thousand.

Line 123: replace the word ‘drew’ with ‘returned’.

Response: Thanks for the suggestion. We have replaced the word ‘drew’ with ‘returned’.

Line 128: methods are typically written in past tense, thus please consider changing the word ‘consists’ to ‘consisted’.

Response: Thanks for the suggestion. We have changed the word ‘consists’ to ‘consisted’.

Line 132: in my experience cuttlefish startle easily, did the cuttlefish exhibit any startle behaviours when the floating container was lowered into the home tank? Had the cuttlefish been habituated to the apparatus and lowering process prior to testing? If yes, please include this in the methods.

Response: Sorry for the confusion. Yes, cuttlefish exhibited some startle behaviors in the first few trials when the experimental apparatus was lowered into the home tank. We had habituated the cuttlefish to the apparatus and lowering process prior to testing. We have added the information of habituation process in the Methods.

Lines 137-140: this sentence is a little confusing. Was the entire apparatus lifted out of the water? If yes, please include an explanation as to why the cuttlefish were not rewarded with the food within the chamber once a choice was made? I assume it is because the authors did not want to reinforce the subject’s choice in this initial stage of the experiment, but this needs to be stated in the methods.

Response: Sorry for the confusion. Yes, the entire apparatus was lifted out of the water in the experiment. We have also made an explanation as to why the cuttlefish were not rewarded with the food within the chamber once a choice was made. Yes, we did not want to reinforce the subject’s decision in the experiment of 1 vs. 2 choice. We have added more detailed information about the experimental procedure in the Methods.

Line 143: The study by *Schnell et al. 2016. Lateralization of eye use in cuttlefish*, is a more appropriate reference to use here because it demonstrates visual lateralization during prey capture, whereas the *Jozet-Alves et al. 2012*. study demonstrates visual lateralization when searching for shelter in a two-choice arm maze.

Response: Thanks for the suggestion. We have replaced the *Jozet-Alves et al 2012* study with the *Schnell et al 2016* study.

Line 163: what exactly constitutes a tiny shrimp? Can you please include parameters?

Response: Sorry for the confusion. A tiny shrimp is of roughly 20% of the cuttlefish’s body length. We have added this information in the Methods.

Lines 182-185: Can the authors please provide a little more detail in the statistical section. Did the authors first conduct normality tests to check parametric assumptions? If yes, please state the type of normality test and the results of that test to justify why non-parametric tests were used.

Response: Sorry for the confusion. Yes, we have first conducted the Kolmogorov-Smirnov normality test to check parametric assumptions. Since some of data tested were not normally distributed, thus we have adopted the non-parametric Wilcoxon Signed Ranks test in the present study. We have stated the type of normality test and the results of that test to justify the use of a non-parametric test in the Methods.

Line 191: consider revising to ‘...cuttlefish innately preferred a larger quantity of prey’.

Response: Thanks for the suggestion. We have revised the sentence.

Line 196: why ‘apparently’? Consider omitting this word.

Response: Thanks for the suggestion. We have omitted the word.

Lines 220-221: this sentence is poorly worded, please consider revising. i.e. ‘in the wild, the prey that cuttlefish feed on are often distributed heterogeneously and different prey types vary significantly in their nutritional value.’

Response: Thanks for the suggestion. We have revised the sentence.

Line 223: omit the word ‘simply’.

Response: Thanks for the suggestion. We have omitted the word.

Line 227: omit the word ‘the’ before ‘prospect theory’; same comment for line 231.

Response: Thanks for the suggestion. We have omitted the words.

Line 232: consider replacing ‘status quo’ with ‘prevailing conditions’.

Response: Thanks for the suggestion. We have replaced ‘status quo’ with ‘prevailing conditions’.

Lines 232-233: due to poor wording, please consider revising the end of sentence to read: ‘for instance, in reference to prevailing conditions or former experiences.’

Response: Thanks for the suggestion. We have revised the sentence.

Lines 233-236: Great example

Response: Thanks for the encouragement.

Line 238: please consider revising this sentence to read: ‘...when cuttlefish did not experience a priming effect, they adopted the...’

Response: Thanks for the suggestion. We have revised the sentence.

Line 246: consider revising sentence to read: ‘the apparent loss aversion effect was prolonged’.

Response: Thanks for the suggestion. However, in this context, we think that it may be better to keep the original sentence “This effect was quite long lasting.”

Line 249: omit the word ‘to’ before ‘dependent’.

Response: Thanks for the suggestion. We have omitted the word.

Line 250: omit the word ‘with’ before ‘six tiny shrimp’; also consider revising sentence to read: ‘when cuttlefish were fed six tiny shrimp without experiencing the priming effect,...’

Response: Thanks for the suggestion. We have omitted the word and revised the sentence.

Line 254: consider replacing the word ‘given’ with ‘prevailing’

Response: Thanks for the suggestion. We have replaced the word ‘given’ with ‘prevailing’.

Lines 255-259: please consider revising this section as the sentence it is poorly worded. For instance, consider breaking the sentence into two parts. Please also consider defining ‘instrumental conditioning experiment’, as a general biological audience that does not study animal cognition might need further explanation.

Response: Thanks for the suggestion. We have revised the sentence. We have also provided an explanation of instrumental conditioning experiment in this section.

Line 259: missing the word ‘of’ after ‘regardless’

Response: Thanks for the suggestion. We have added the word ‘of’ after ‘regardless’.

Lines 259-261: please consider revising the sentence. Currently, it’s a little confusing. What exactly is helping the cuttlefish optimise their foraging strategy? Reference-dependent evaluation or value-based decision making or both?

Response: Sorry for the confusion. We have revised the sentence by removing “value-based decision-making process”.

Lines 267-270: please consider revising the last sentence. The statement requires expanding and further justification. Why is foraging cognition in cuttlefish equivalent to that of vertebrates? What specific aspects are equivalent? Are the authors referring to the previous statement in that state-dependent valuation learning is rife across diverse taxa? If yes, then foraging cognition in cuttlefish is not only akin to vertebrates but also other invertebrates i.e., insects. If no, what specific aspects of learned valuation is vertebrate-like? Why is this important?

Response: Sorry for the confusion. To avoid the ambiguity in the comparison of foraging cognition across diverse taxa, we have deleted the last paragraph of Discussion completely.

Lines 365-373: Please revise the figure 2 legend to clearly state that cuttlefish in the experiment depicted in 2D did not get exposed to a priming effect.

Response: Thanks for the suggestion. We have revised the figure legend.

Appendix B

RS0S-201602: Learned valuation during forage decision making in cuttlefish

The paper reads well, and all my previous queries have been resolved. I also acknowledge that changes requested by the other referee and the editor have improved the manuscript. I have some minor comments below but overall this is a job well done!

Minor comments

Line 40: consider removing the word ‘apparently’

Line 41: I think there is a word missing before ‘fed’, please insert ‘were’

Line 44: consider replacing the term ‘elevated taking’ with ‘preference’

Line 142-143: I’m not sure this second sentence is needed, are you not just repeating what you have written just prior?

Line 150: insert ‘the’ between during and experiment so that the sentence reads ‘... were vigorously active during the experiment, the...’;

Line 158: if you are referring to single cuttlefish in this sentence, which by the looks of the apostrophe you are, then you are missing the article ‘the’ before ‘cuttlefish’s foraging decision’. Same comment for **line 179** and **line 213**.

Line 189-192: Can you please rephrase this sentence because in its current form it is a little difficult follow. For example, ‘to check parametric assumptions we used X test. Results revealed that our data was not normal (results of X test). We thus proceeded with a Wilcoxon Signed Ranks test.

Line 202: this should be in past tense, replaced ‘influences’ with ‘influenced’

Line 222: ‘prefers’ should be singular i.e. ‘prefer’

Line 268: please rephrase for e.g. in the prawn-in-the-tube test, cuttlefish learn to stop attacking the prawn, and the level...

Line 271: modified not modifiable

Great work 😊

Appendix C

RS0S-201602: Learned valuation during forage decision making in cuttlefish

Reviewer: 1

The manuscript has improved substantially. This new version avoids unnecessary claims of higher cognitive processes. I still fail to understand how the results found can be interpreted using prospect theory (page 12). The fact that decision-making is based on relative rather than on absolute metrics does not help us explain how 1 shrimp is better than 2. (by the way, a phenomenon cannot be akin to a theory, lines 233-236). Positive reinforcement is indeed a good and parsimonious explanation.

The pending issue is whether the demonstration of reinforcement learning in cuttlefish is newsworthy. I do not think it is, but the science presented in the paper is solid.

Response: Thanks for the encouragement and the reminder of parsimonious explanation. As we explained in the last response to reviewer, the most likely interpretation of our result is indeed reinforcement learning, but there is still a possibility that the concept of prospect theory may be applied. The fact that decision-making is based on relative rather than on absolute metrics and the cuttlefish's choice is determined relative to a reference point, such as a former experience (the outcome of 0 vs. 1 choice), is a potential example of prospect theory. We have toned down our claim further by adding "This phenomenon **may be** akin to prospect theory", and we hope that reviewer will allow us to present this alternative possibility in the revised manuscript, so that the readers can have a chance to evaluate this interpretation by themselves. Indeed, the demonstration of reinforcement learning in cuttlefish may not be a novel finding in itself. However, reinforcement learning in the context of choosing between different quantity of prey has a significant implication in cuttlefish's foraging strategy and behavioral ecology.

Reviewer: 2

The paper reads well, and all my previous queries have been resolved. I also acknowledge that changes requested by the other referee and the editor have improved the manuscript. I have some minor comments below but overall this is a job well done!

Response: Thanks for the encouragement.

Minor comments

Line 40: consider removing the word 'apparently'

Response: Thanks for the suggestion. We have omitted the word.

Line 41: I think there is a word missing before 'fed', please insert 'were'

Response: Thanks for the suggestion. However, since this is a noun clause, we think that 'were' is not needed in here. If the editor suggests to add this word, we will insert it at the final editing stage.

Line 44: consider replacing the term 'elevated taking' with 'preference'

Response: Thanks for the suggestion. We have replaced the term 'elevated taking' with 'preference'.

Line 142-143: I'm not sure this second sentence is needed, are you not just repeating what you have written just prior?

Response: Sorry for the confusion. The first sentence is used to emphasize the point in time when the reward is given. The second sentence is used to explain the source of the reward, i.e., to emphasize that the reward was not from the shrimp in the apparatus during experiment.

Line 150: insert 'the' between during and experiment so that the sentence reads '... were vigorously active during the experiment, the...';

Response: Thanks for the suggestion. We have inserted the word 'the' between during and experiment.

Line 158: if you are referring to single cuttlefish in this sentence, which by the looks of the apostrophe you are, then you are missing the article ‘the’ before ‘cuttlefish’s foraging decision’. Same comment for **line 179** and **line 213**.

Response: Thanks for the suggestion. We have inserted the word ‘the’ in line 158, line 179 and line 213.

Line 189-192: Can you please rephrase this sentence because in its current form it is a little difficult follow. For example, ‘to check parametric assumptions we used X test. Results revealed that our data was not normal (results of X test). We thus proceeded with a Wilcoxon Signed Ranks test.

Response: Thanks for the suggestion. We have rephrased this sentence.

Line 202: this should be in past tense, replaced ‘influences’ with ‘influenced’

Response: Thanks for the suggestion. We have replaced ‘influences’ with ‘influenced’.

Line 222: ‘prefers’ should be singular i.e. ‘prefer’

Response: Thanks for the suggestion. We have replaced ‘prefers’ with ‘prefer’.

Line 268: please rephrase for e.g. in the prawn-in-the-tube test, cuttlefish learn to stop attacking the prawn, and the level...

Response: Thanks for the suggestion. We have rephrased this sentence.

Line 271: modified not modifiable

Response: Thanks for the suggestion. We have replaced ‘modifiable’ with ‘modified’.

Great work

Response: Thanks for the encouragement.